# Revisiting the factor structure of the Short-Form McGill Pain Questionnaire-2 (SF-MPQ-2): Evidence for a bifactor model in individuals with Chiari malformation

**David M. Tokar[1] \*, Kevin P. Kaut[2], Philip A. Allen[2]**

**1** Department of Social and Behavioral Sciences, Central State University, Wilberforce, OH, United States of America, **2** Department of Psychology, University of Akron, Akron, OH, United States of America

\* dtokar@centralstate.edu

**Data Availability Statement:** All relevant data are within the paper and its Supporting Information files.

## Abstract

The Short-Form McGill Pain Questionnaire-2 (SF-MPQ-2; Dworkin et al., 2009) is intended to measure the multidimensional qualities of pain (i.e., continuous, intermittent, neuropathic, and affective) as well as total pain. Using structural equation modeling, we evaluated the fit of four competing measurement models of the SF-MPQ-2—an oblique 4-factor model, a 1-factor model, a higher-order model, and a bifactor model—in 552 adults diagnosed with Chiari malformation, a chronic health condition whose primary symptoms include head and neck pain. Results revealed the strongest support for the bifactor model, suggesting that SF-MPQ-2 item responses are due to both a general pain factor and a specific pain factor that is orthogonal to the general pain factor. Additional bifactor analyses of the SF-MPQ-2's model-based reliability and dimensionality revealed that most of the SF-MPQ-2's reliable variance is explained by a general pain factor, and that the instrument can be modeled unidimensionally and scored as a general pain measure. Results also indicated that the general and affective pain factors in the bifactor model uniquely predicted pain-related external criteria (e.g., depression, anxiety, and stress); however, the continuous, intermittent, and neuropathic factors did not.

## Introduction

In the 2019 National Health Interview Survey, 50.2 million adults (20.5%) claimed that they experienced pain on most or every day [1–3], and this estimate will undoubtedly increase as the population continues to age [3]. Given the tremendous impact of chronic pain on individuals and society, it is not surprising that chronic pain has become a topic of increasing interest to psychologists [4]. Since the 1965 publication of Melzack and Wall's seminal gate control theory of pain, [5] considerable research has supported the notion that the chronic pain experience is subjective and multidimensional, consisting of biological, psychosocial, and behavioral components [4,6–8].

**Funding:** This research was supported by a Conquer Chiari Research Grant from the Conquer Chiari Foundation as well as NIH grant 1R15NS109957-01A1. The funders did not play any role in the study design, data collection and analysis, decision to publish, or preparation of this manuscript.

**Competing interests:** No–we have no competing interests.

Clearly, reliable and valid assessment tools are required for the systematic investigation, accurate classification, and effective treatment of chronic pain conditions [6,7]. Over the past 40 years, the McGill Pain Questionnaire (MPQ [9]) and the Short-Form McGill Pain Questionnaire (SF-MPQ [10]) have been among the most widely used self-report measures of pain qualities. Research has tended to support the reliability and validity of the different versions of the MPQ, including the most recently revised and expanded version of the SF-MPQ (SF-MPQ-2; Dworkin et al., 2009 [11]); however, the evidence regarding the factor structure of the SF-MPQ-2 is equivocal.

The organization of the SF-MPQ-2 underscores the multidimensional perspective of pain, with items clustered into four subscales reflecting the subjectively experienced qualities of pain as *continuous* (e.g., "throbbing pain"; "cramping pain"; "gnawing pain"), *intermittent* (e.g., "shooting pain"; "sharp pain"; "stabbing pain"), *affective* (e.g., "tiring-exhausting"; "punishing"; "fearful"), and *neuropathic* (e.g., "hot-burning pain"; "cold-freezing pain"; "itching pain"). This measure provides users with an efficient assessment of these pain dimensions, as well as total pain. As such, it is likely to be the instrument of choice in future clinical research. The primary purpose of the current study was to evaluate further the factor structure of the SF-MPQ-2 in a large sample of individuals diagnosed with a chronic health condition whose primary symptoms manifest as head and neck pain (i.e., Chiari malformation; see Method for clinical description).

It should be noted that Dworkin et al. (2009) [11] developed the SF-MPQ-2 in order to assess the growing chronic health concerns associated with 'neuropathic pain'. To some extent, the emergence of neuropathic pain as a significant and chronic health issue reflects evolving societal health concerns and outcomes, coupled with advances in pain research and diagnostic specificity [12]. Causes of neuropathic pain (i.e., defined as a disease or lesion of the somatosensory system) likely reflect, in part, sequelae associated with an aging population, including the incidence of diabetes, neurodegenerative conditions (e.g., Parkinson's disease), and stroke; in addition, diverse nervous system pathologies such as post-surgical pain, spinal cord injury, and multiple sclerosis are included here, as well as the effects of other diseases such as cancer and HIV infection (see Colloca et al., 2017, for comprehensive review) [12].

Dworkin et al. [11] specifically addressed the issue of neuropathic pain in their development of the SF-MPQ-2 by adding nine items to the original 15 of the SF-MPQ. Using a web-based sample of 882 chronic pain patients (i.e., neuropathic pain, $n = 349$, non-neuropathic pain, $n = 533$), exploratory factor analyses (EFAs) on 20 sensory pain items (i.e., four affective pain items were not included, but were retained for the final version of the SF-MPQ-2) resulted in a three-factor structure that was generally invariant across the neuropathic pain and non-neuropathic pain groups. After dropping two items loading only moderately on the second factor, EFAs on the remaining 18 sensory pain items again yielded a three-factor structure that was generally invariant across the neuropathic pain and non-neuropathic pain groups. Based on their EFA results and prior research on symptoms of neuropathic and non-neuropathic pain, the four subscales comprising the SF-MPQ-2 were established (i.e., continuous, intermittent, neuropathic, and affective).

Psychometric properties of the SF-MPQ-2 have been respectable, with internal consistency reliabilities ($\alpha$s) ranging from .73 to .87 for the four pain subscale scores and $\alpha$s of .91 and .95 for total pain scores in two separate samples [11]. Construct validity has been documented via positive correlations with measures of pain intensity and impact. In addition, Dworkin et al. (2009) [11] utilized confirmatory factor analysis (CFA) to evaluate the fit of their web-based survey data to a four-factor structure consistent with the scoring of the SF-MPQ-2's four pain subscales. However, and of particular relevance to the present study, rather than simultaneously modeling all four pain factors and allowing them to covary, Dworkin et al. (2009) [11]

performed separate CFAs on the items composing each of the four respective SF-MPQ-2 subscales. This forced restriction of covariance among items across subscales is potentially problematic, reiterating the question of whether a four-factor model adequately fits the SF-MPQ-2 data, or a more parsimonious (e.g., one-factor, higher-order) model better represents the variability in SF-MPQ-2 item responses.

Subsequent psychometric investigations of the SF-MPQ-2 have been illustrative. Using patient samples with diverse pain conditions has yielded consistent support for the internal consistency and convergent validity of the specific pain subscales as well as total pain scale scores [13–15]. However, independent tests of the SF-MPQ-2's factor structure have yielded equivocal findings. Gauthier et al. (2014) [14], using a combined sample of 190 older and younger patients with cancer pain, tested an oblique four-factor model corresponding to the four intended SF-MPQ-2 pain constructs and found a "reasonable fit" (p. 763) that was shown to be configurally invariant across the two age groups. Yet, several CFA fit index values (e.g., CFI = .78, TLI = .75) indicated a poor fit to the data. Moreover, latent factor correlations were high (median $r$s of .71 and .85 in the younger and older groups, respectively), suggesting that a higher-order structure, in which the first-order factors load onto a superordinate general pain factor, might provide a better fit. Dworkin et al. (2015) used CFA to test the fit of a four-factor model corresponding to the SF-MPQ-2's four intended pain constructs to data obtained from 666 acute lower back pain patients. Results indicated a good fit for the Continuous subscale data, but mixed or no support for the Intermittent, Neuropathic, and Affective subscale data. Like Dworkin et al. (2009) [11], Dworkin et al. (2015) [13] performed separate CFAs on each set of items corresponding to a specific SF-MPQ-2 pain construct instead of testing the dimensionality of the total measure. As such, conclusions regarding the structure of the SF-MPQ-2 remain equivocal, and certainly underscore the need for continued efforts to assess the psychometric properties of the scale.

In perhaps the most comprehensive examination of the SF-MPQ-2's factor structure, Lovejoy et al. (2012) [15] evaluated the fit to their data ($N$ = 186 veterans with chronic pain symptoms) of three different structural models. The first model was an oblique four-factor model corresponding to the SF-MPQ-2's four intended pain constructs. The second model was a one-factor model corresponding to a total pain construct in which all 22 SF-MPQ-2 items loaded onto one general factor. The third model was a higher-order model in which the SF-MPQ-2 items loaded onto one of four domain-specific pain factors, which in turn contributed to a higher-order general pain factor. CFA results revealed the strongest support for the oblique four-factor and higher-order models. Support for the oblique four-factor model implies four related but distinct pain constructs. Support for the higher-order model implies a hierarchical structure in which a general pain factor representing the shared variance among the first-order pain factors ultimately (i.e., and indirectly via the first-order factors) accounts for variation in SF-MPQ-2 item responses. That is, support for the higher-order model implies that the SF-MPQ-2 item responses contain variance attributable to specific pain constructs (e.g., continuous, intermittent, neuropathic, affective) as well as a higher-order general pain construct [16].

Although Lovejoy et al.'s (2012) [15] CFA results suggest the possibility that an oblique four-factor model or a higher-order model best represents the structure of the SF-MPQ-2, their findings must be interpreted cautiously for at least two reasons. First, neither the oblique four-factor model nor the higher-order model met recommended fit index value cutoffs for adequate fit (e.g., CFI close to .95, RMSEA $\leq$ .06; Kline, 2016) [17]. Second, direct (e.g., $\chi^2$ difference) tests comparing the fit of the different nested models were not performed; therefore, the relative superiority of the oblique four-factor and higher-order models is questionable. Thus, a primary purpose of the present study is to evaluate the absolute and relative fit of an

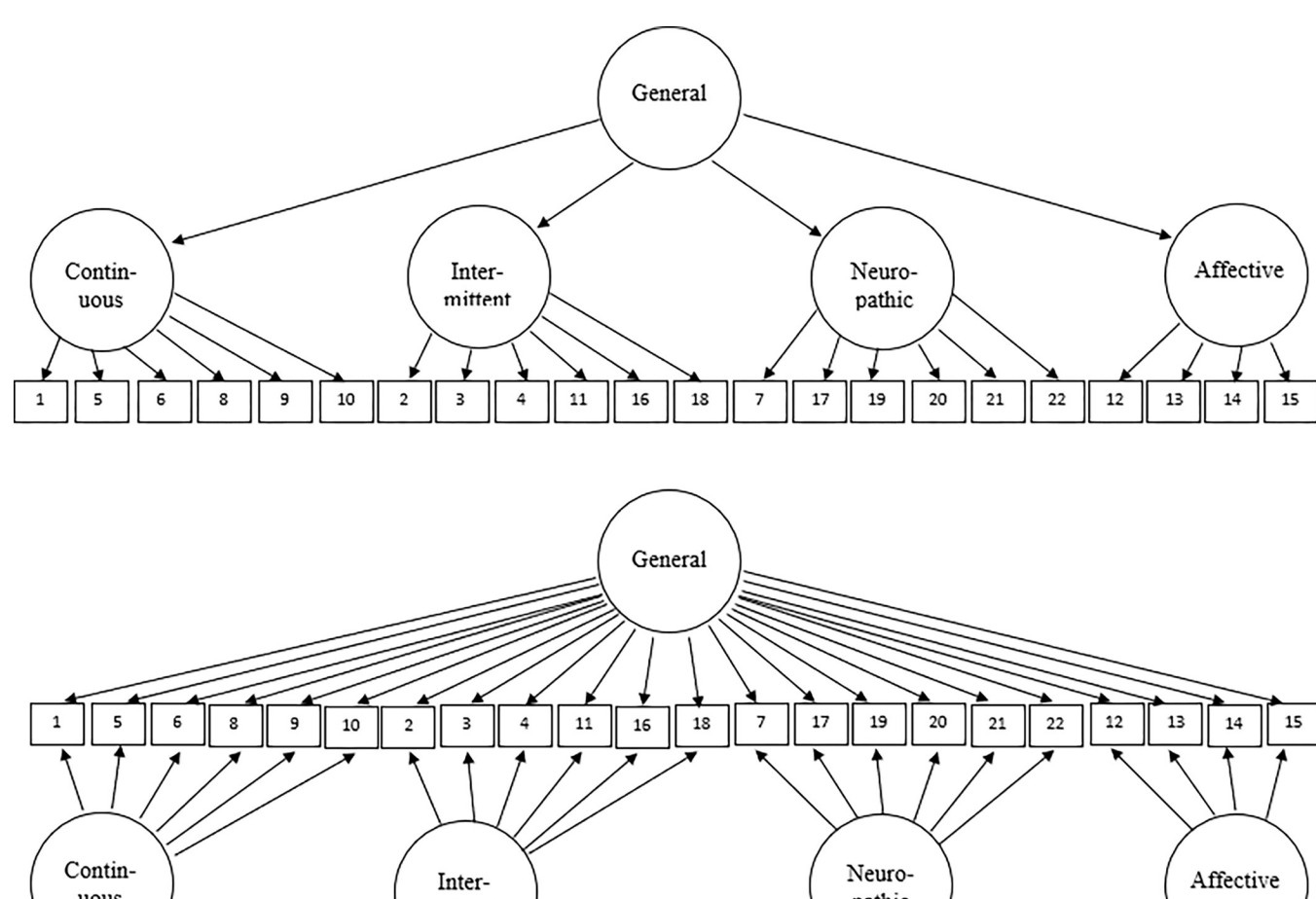

**Fig 1. Higher-order model depicted on the top.** Bifactor model depicted on the bottom. SF-MPQ-2 item numbers are presented in the rectangles.

oblique four-factor model, a one-factor general model, and a higher-order model (see Fig 1) for SF-MPQ-2 data in a large sample (*N* = 552) of patients diagnosed with Chiari malformation, a condition associated with skeletal abnormalities (often congenital in nature), and commonly presenting with chronic pain and related health complications (see the Participants section below for a more detailed description of Chiari malformation).

In our work, we also considered an alternative bifactor model, which has yet to be evaluated with the SF-MPQ-2. In contrast to the higher-order model, bifactor models assume two independent sources of common variance directly influencing all of the SF-MPQ-2 item responses (see Fig 1). That is, bifactor models imply that item responses are potentially influenced by both a general factor (e.g., overall pain) and a specific factor (i.e., one of the four pain dimensions of the SF-MPQ-2) that is orthogonal to the general factor and the other specific factors. Estimates from bifactor models enable researchers to determine whether domain-specific factors explain variability in item responses independent of a general latent factor. Thus, bifactor models can be used to help clarify the dimensionality of a measure and determine whether scoring an instrument for raw total and raw subscale scores is warranted [18]. Assuming a bifactor model best represents the structure of the SF-MPQ-2, additional bifactor analyses (e.g., model-based reliability, explained common variance; see Rodriguez et al., 2016a, 2016b)

[18,19] could shed light on the appropriateness of scoring and interpreting the instrument for four domain-specific pain constructs [11,14,15] and for a general total pain construct [11,15].

Finally, assuming the SF-MPQ-2 structure is best represented by a bifactor model, we investigated the incremental validity of the SF-MPQ-2 general and specific factors (e.g., continuous pain) by examining their unique relations with several external criteria (i.e., symptom severity, neck pain, depression, anxiety, and stress). Consistent with previous research demonstrating moderate to strong positive associations of sensory (i.e., continuous, intermittent, and neuropathic), affective, and total pain scores with pain severity [15] and intensity [13,14], we hypothesize that continuous, intermittent, neuropathic, affective, and general pain factor scores will relate uniquely and positively with symptom severity and neck pain. Based on previously reported positive associations of anxiety and depression with all four types of (and total) pain [13,15], we hypothesize that continuous, intermittent, neuropathic, affective, and general pain factor scores will relate uniquely and positively with anxiety, depression, and stress. Because the construct reliability of the general pain factor is likely to be higher than that of the "residualized" specific pain factors (Rodriguez et al., 2016b, p. 146) [18], we anticipate the strongest unique associations between the general pain factor and the external criteria.

## Method

### Ethics statement

The research reported in the present paper was approved by the Institutional Review Board at The University of Akron. All participants provided electronic informed consent.

### Participants and procedure

We collected data using a web-based survey containing the SF-MPQ-2 and other measures described below. We recruited a sample of 552 adult participants (i.e., $\geq$ 18 years of age) from the Chiari 1000 database (i.e., a database established for the collection of behavioral and anatomical information from Chiari malformation patients; see Allen et al., 2018) [20] who had been diagnosed with Chiari malformation (CM). This malformation is often congenital in nature, impacting cranial and spinal column anatomy, and typically involving the base of the skull, upper vertebral column, and associated neural tissue [21]. In particular, the radiological definition of CM specifies this anatomical issue, involving portions of the cerebellum (i.e., cerebellar tonsils) extending at least 5 mm below the line of demarcation between the brain and the spinal cord—an opening in the base of the skull known as the foreman magnum [22]. Etiological, diagnostic, and treatment considerations continue to emerge, particularly in the context of modern neuroimaging advances [21,23,24]. Not uncommonly, reduced cerebrospinal fluid volume in the posterior portions of the skull (i.e., posterior cranial fossa, beneath the occipital lobes) and the anterior cerebrospinal fluid (CSF) space in the upper cervical region of the spinal canal exacerbate cervico-medullary compression by the cerebellum, thereby exerting pressure (i.e., particularly during diastolic cardiac cycles) on underlying brainstem and cervical spine regions [21,25,26]; see also [24]. Also, pain levels (as assessed by the SF-MPQ-2) in CM relative to healthy age- and education-matched controls are associated with significantly different levels of white-matter integrity as indexed by diffusion tensor imaging (DTI, Houston et al., 2020 [27]) as well as intrinsic functional connectivity as indexed via resting-state functional magnetic resonance imaging (fMRI) (Houston et al., 2021 [28]).

Variations in the degree of structural abnormality naturally influence functional and sensory outcomes, although the most commonly experienced physical symptoms include chronic pain, headache, and sensory-motor disturbances typically affecting regions of the head, neck,

and upper extremities [21]. Increased levels of anxiety and depression are also observed in CM [29] (Garcia et al., 2019), as well as cognitive dysfunction [29–33].

In the present sample, the most frequently reported comorbid illnesses were autoimmune disease (40.22%), scoliosis (23.95%), syringomyelia (22.85%), pseudotumors (10.24%), and Ehlers-Danlos syndromes (9.69%). Demographic data were based on 548 (99.3% of the total sample) cases for whom such data were available. Participants were 518 (94.5%) women and 30 (5.5%) men who ranged in age from 18 to 66 years ($M$ = 37.80, $SD$ = 10.92). The majority (88.1%) of participants identified ethnically as White/European American, with 4.6% Black/African American, 4.2% Hispanic, 2.4% American Indian/Native American, 0.5% Asian/Asian American, and 0.2% Pacific Islander. In terms of employment status, 53.5% indicated that they were not currently working, whereas 46.5% indicated that they were currently working. Regarding highest level of education completed, 27.0% indicated some college, 20.8% high school graduate, 15.3% bachelor's degree, 13.5% associate's degree, 10.9% trade school/technical school, 10.9% master's degree, and 1.5% doctorate degree.

All participants scored a minimum of 1 or above (i.e., indicating at least some degree of pain intensity) on one or more of the four SF-MPQ-2 pain subscales (i.e., continuous, intermittent, neuropathic, or affective). Chiari malformation is not considered to be a neuropathic disorder (i.e., involving disease or lesion of the somatosensory system; see Treede et al., 2008 [34]); therefore, we assume that our sample is primarily non-neuropathic in nature. Institutional Review Board approval was obtained before the onset of this study, and all participants consented before they participated.

## Measures

In addition to the SF-MPQ-2, we collected data on the Neck Disability Index (Vernon & Mior, 1991 [35]), self-rated Chiari symptom severity, and the Depression Anxiety Stress Scales-21 (DASS-21; Lovibond & Lovibond, 1995 [36]).

## SF-MPQ-2

The SF-MPQ-2 [11] is a 22-item self-report measure of different pain qualities or related symptoms. Participants used an 11-point Likert scale (0 = *none*, 10 = *worst possible*) to indicate the intensity of each pain quality/symptom experienced within the past week. The SF-MPQ-2 is scored for continuous (6 items; e.g., "throbbing"), intermittent (6 items; e.g., "shooting"), neuropathic (6 items; e.g., "tingling"), and affective (4 items; e.g., "tiring-exhausting") pain as well as total pain by calculating the mean for each subscale (or total), with higher scores corresponding to more intense pain symptoms. Dworkin et al. (2009) reported Cronbach's alphas of .91 (total pain; current $\alpha$ = .94), .73 (Continuous pain; current $\alpha$ = .82), .85 (Intermittent pain; current $\alpha$ = .89), .78 (Neuropathic pain; current $\alpha$ = .83) and .77 (Affective pain; current $\alpha$ = .82) in a sample of 882 adults with chronic pain. Gauthier et al. (2015) [14] demonstrated support for the convergent validity of the SF-MPQ-2 through expected relations with measures of pain intensity, interference, and relief; depressive symptoms; and physical and mental health quality of life.

**Neck Disability Index.** We used the Neck Disability Index [35] (NDI), an adapted version of the Oswestry Low Back Pain Disability Questionnaire (Fairbanks, Couper, Davies, & O'Brien, 1980) [37], to assess neck pain disability, a common symptom of Chiari malformation [38]. Participants used a 6-point Likert scale (0 = no interference [e.g., "I can do as much work as I want to"], 5 = extreme interference [e.g., "I cannot do any work at all"]) to indicate the extent to which their neck pain has affected their ability to perform eight different everyday activities (e.g., working, driving), as well as their experience of pain intensity (0 = "I have no

pain at the moment," 5 = "The pain is the worst imaginable at the moment") and headaches (0 = "I have no headaches at all," 5 = "I have headaches almost all the time"). Scores for each of the 10 items are summed and then doubled; thus, total scores can range from 0–100, with higher scores indicating greater disability. McCarthy, Grevitt, Silcocks, and Hobbs (2007) [39] reported a Cronbach's alpha of .86 (current $\alpha$ = .89) and demonstrated support for the concurrent validity of the NDI through expected relations with the Short Form 36 Health Survey Questionnaire (SF36; Brazier et al., 1992) [40], a measure of functional ability and overall health and well-being, in a sample of 160 patients with neck pain.

**Symptom severity.** Participants used a 4-point Likert-scale (1 = *mild*, 4 = *very severe*) to indicate their overall level of Chiari malformation symptom severity, which is characterized by chronic headache and neck pain [38], anxiety and depression [29], and cognitive dysfunction [30–33].

**Depression Anxiety Stress Scales-21.** We used the 21-item Depression Anxiety Stress Scales-21 (DASS-21) [36] to measure self-reported levels of depression, anxiety, and stress. Each of the three DASS-21 subscales is composed of seven items. Participants used a 4-point Likert scale (0 = *did not apply to me at all*; 3 = *applied to me very much*, *or most of the time*) to indicate how frequently they experienced each within the past week. Sample items include "I couldn't seem to experience any positive feeling at all" (Depression), "I felt scared without any good reason" (Anxiety), and "I found it hard to wind down" (Stress). Scores for each DASS-21 subscale are summed and multiplied by two (for comparability with the DASS-42), with higher scores corresponding to higher levels of each construct. Page, Hooke, and Morrison (2007) [41] reported Cronbach's alphas of .96 (Depression; current $\alpha$ = .92), .92 (Anxiety; current $\alpha$ = .83), and .95 (Stress; current $\alpha$ = .85) in a sample of 124 adult patients diagnosed with depression. Henry and Crawford (2005) demonstrated support for the convergent validity of the DASS-21 via expected relations with independent measures of depression and anxiety.

## Results

### Missing data

Seventy-three of the 552 cases contained at least one missing item-level data point, and the overall rate of missing data was 6.18%. Little's (1988) [42] missing completely at random (MCAR) test was nonsignificant, $\chi^2$ (67) = 52.68, $p$ = .90, indicating that missingness was not systematically related to any of the study variables. We used full information maximum likelihood (FIML) estimation to deal with missing data in all structural equation modeling (SEM) analyses. FIML uses all available data to generate unbiased parameter estimates and standard errors (Tabachnick & Fidell, 2013) [43].

### Descriptive statistics and intercorrelations

Prior to conducting structural equation modeling (SEM) analyses, all measured variables were evaluated for normality and outliers. All variables satisfied assumptions of univariate normality (i.e., absolute skewness and kurtosis values $\leq$ 2; Tabachnick & Fidell, 2013) [43], and no univariate outliers were identified. However, Mardia's (1970) [44] tests for multivariate skewness and kurtosis indicated violations of multivariate normality; therefore, all SEM analyses were performed using a robust estimator (i.e., MLR) to correct for multivariate non-normality (Muthén & Muthén, 1998–2017) [45].

Means, standard deviations, alpha reliabilities, and intercorrelations of all measured variables are reported in Table 1. SF-MPQ-2 total and subscale score means were comparable (i.e., within roughly half a standard deviation) to those reported by Dworkin et al. (2009) [11] based on samples of chronic pain patients. SF-MPQ-2 alphas (ranging from .82 to .89 for subscales, .94 for total pain scores) were comparable to those reported in other chronic pain patient

**Table 1. Intercorrelations, internal consistencies, means, and standard deviations of all variables.**

| Variable | 1 | 2 | 3 | 4 | 5 | 6 | 7 | 8 | 9 | 10 |
|---|---|---|---|---|---|---|---|---|---|---|
| 1. Total Pain | – | | | | | | | | | |
| 2. Continuous Pain | .90 | – | | | | | | | | |
| 3. Intermittent Pain | .91 | .74 | – | | | | | | | |
| 4. Neuropathic Pain | .88 | .70 | .73 | – | | | | | | |
| 5. Affective Pain | .85 | .73 | .69 | .64 | – | | | | | |
| 6. Neck Pain | .64 | .58 | .61 | .52 | .53 | – | | | | |
| 7. Symptom Severity | .20 | .16 | .21 | .13 | .19 | .26 | – | | | |
| 8. Depression | .38 | .31 | .35 | .28 | .43 | .42 | .15 | – | | |
| 9. Anxiety | .48 | .42 | .40 | .41 | .50 | .48 | .16 | .68 | – | |
| 10. Stress | .39 | .33 | .33 | .30 | .45 | .34 | .09 | .68 | .73 | – |
| $\alpha$ | .94 | .82 | .89 | .83 | .82 | .89 | NA | .92 | .83 | .85 |
| $M$ | 4.16 | 4.61 | 3.95 | 3.82 | 4.33 | 48.05 | 2.73 | 13.31 | 13.16 | 17.37 |
| $SD$ | 2.26 | 2.38 | 2.75 | 2.43 | 2.71 | 18.41 | 0.85 | 11.15 | 10.14 | 10.13 |

*Note. Ns* ranged from 479–552. All correlations are significant at $p < .05$.

samples (Dworkin et al., 2009 [11]; Lovejoy et al., 2012 [15]). SF-MPQ-2 subscale intercorrelations were large (median $r = .72$), and correlations between total and subscale scores (*mdn* = .89) approached unity, suggesting that the subscales may not be measuring unique pain constructs. Associations of SF-MPQ-2 total and subscale scores with measures of neck pain, symptom severity, depression, anxiety, and stress were positive and (except for those involving symptom severity) of moderate to large magnitude.

### SP-MPQ-2 factor structure

We performed SEM analyses using Mplus version 8.1 (Muthén & Muthén, 1998–2018) [45] to examine the fit of four different structural models (described below) to our data. Model fit was evaluated using the Satorra-Bentler scaled (i.e., mean-adjusted) $\chi^2$ goodness-of-fit test, comparative fit index (CFI), root mean square error of approximation (RMSEA), and standardized root mean square residual (SRMR). CFI values close to .95, RMSEA values $\leq .06$, and SRMR values $\leq .08$ indicate a good fit (Kline, 2016) [17]. We used the Satotta-Bentler scaled $\chi^2$ difference test to compare the goodness-of-fit of the different nested models and differences in the other fit index values to compare nonnested models.

**Model 1: Oblique four-factor model.** The first model was an oblique four-factor model based on Dworkin et al.'s (2009) recommended scoring of the SF-MPQ-2 [11]. Continuous, intermittent, neuropathic, and affective pain constructs were modeled as correlated latent factors. Items scored for each SF-MPQ-2 subscale were indicators of the corresponding latent factor. As shown in Table 2, fit index values indicated that Model 1 did not fit the data well.

**Model 2: One-factor model.** The second model was a one-factor general pain model with all 22 SF-MPQ-2 items as indicators of a single latent factor. As shown in Table 2, Model 2 had a poor fit to the data. Relative to Model 1, Model 2 provided a significantly worse fit, $\Delta \chi^2$ (6, $N = 552$) = 178.35, $p < .001$.

**Model 3: Higher-order model.** The third model was a higher-order model with four first-order pain factors identical to those in Model 1 as well as a higher-order general pain factor onto which the first-order factors loaded. As shown in Table 2, Model 3 did not fit the data well. However, Model 3 was not a significantly worse fit than Model 1, $\Delta \chi^2$ (2, $N = 552$) = 5.94, $p > .05$.

**Table 2. Summary of fit statistics for all models.**

| Model Description | $\chi^2$ | $df$ | CFI | RMSEA | SRMR |
|---|---|---|---|---|---|
| Model 1: Oblique four-factor | 907.38 | 203 | .87 | .079 | .067 |
| Model 2: One-factor | 1258.44 | 209 | .81 | .095 | .057 |
| Model 3: Higher-order | 913.27 | 205 | .87 | .079 | .068 |
| Model 4: Bifactor | 458.38 | 187 | .95 | .051 | .038 |

*Note. N* = 552. *df* = degrees of freedom; CFI = comparative fit index; RMSEA = root mean square error of approximation
SRMR = standardized root mean square residual.

**Model 4: Bifactor model.** The fourth model was a bifactor model in which the indicator variables (i.e., items) had loadings on both a specific pain factor corresponding to one of the four factors in Model 1 and a general pain factor. All factor covariances were fixed to zero. Although the $\chi^2$ value was statistically significant (indicating a non-perfect fit), the other fit index values indicated that Model 4 provided a good fit to the data (see Table 2). Furthermore, Model 4 provided a significantly better fit than did the more parsimonious higher-order model (Model 3), $\Delta\chi^2$ (18, $N$ = 552) = 479.21 $p$ < .001. Overall, results indicated that the bifactor model was the best-fitting model in our data.

As shown in Table 3, all 22 items loaded significantly and substantively (ranging from .47 to .75, *mdn* = .65) on the general pain factor. Conversely, only 12 of 22 items loaded significantly on a specific pain factor, and those factor loadings varied considerably in terms of magnitude. For example, the four significant neuropathic pain factor loadings were .11 (item 19), .14 (item 20), .60 (item 22), and .77 (item 21). Furthermore, general pain factor loadings equaled or exceeded specific pain factor loadings for 21 of the 22 items.

Factor loadings in bold are significant at $p$ < .05.

## Model-based reliability and dimensionality

Next, we calculated a number of additional bifactor indices to evaluate the reliability of the SF-MPQ-2 total and subscale pain scores as well as the dimensionality of the SF-MPQ-2. Model-based estimates of internal consistency of the total and subscale scores were calculated using coefficient omega and coefficient omega hierarchical. Omega ($\omega$) and omega subscale ($\omega_S$) reflect the proportion of variance in total scores and subscale scores, respectively, accounted for by all common variance sources (i.e., the general factor and corresponding specific factor[s]) (Rodriguez et al., 2016a, 2016b) [18,19]. Omega hierarchical ($\omega_H$) reflects the proportion of total score variance explained by the general factor after accounting for the variance attributed to the specific factors, whereas omega hierarchical subscale ($\omega_{HS}$) reflects the proportion of subscale score variance explained by the corresponding specific factor after removing the variance attributed to the general factor (Rodriguez et al., 2016a, 2016b) [18,19]. High $\omega_H$ values suggest that total scores are due primarily to a general factor, whereas high $\omega_{HS}$ values suggest that subscale scores are due primarily to a specific factor common to the items composing that subscale. Although there are no clear cutoffs for evaluating $\omega_H$ and $\omega_{HS}$, Reise, Bonifay, and Haviland (2013) [46] recommended "that a minimum would be greater than .50, and values closer to .75 would be much preferred" (p. 137) for determining the unique information provided by total and subscale scores.

Omega coefficients for the total ($\omega$ = .96) and specific pain scores ($\omega_S$ range = .84-.91) were high (see Table 3), indicating that the total and subscale composite scores were highly reliable.

**Table 3. Factor loadings for unidimensional and bifactor solutions.**

| SF-MPQ-2 item | 1-factor | Bifactor Model | | | | |
| --- | --- | --- | --- | --- | --- | --- |
| | | Gen. | Contin. | Intermit. | Neuro. | Affect. |
| 1 (throbbing) | .64 | .64 | .09 | | | |
| 5 (cramping) | .59 | .60 | .06 | | | |
| 6 (gnawing) | .57 | .59 | .06 | | | |
| 8 (aching) | .64 | .62 | .58 | | | |
| 9 (heavy) | .71 | .70 | .28 | | | |
| 10 (tender) | .64 | .65 | .12 | | | |
| 2 (shooting) | .75 | .70 | | .42 | | |
| 3 (stabbing) | .77 | .71 | | .62 | | |
| 4 (sharp) | .77 | .71 | | .49 | | |
| 11 (splitting) | .75 | .75 | | .08 | | |
| 16 (electric-shock) | .63 | .63 | | .07 | | |
| 18 (piercing) | .73 | .74 | | .08 | | |
| 7 (hot-burning) | .65 | .66 | | | .07 | |
| 17 (cold-freezing) | .57 | .59 | | | .06 | |
| 19 (caused by lt. touch) | .66 | .66 | | | .11 | |
| 20 (itching) | .48 | .47 | | | .14 | |
| 21 (tingling) | .66 | .62 | | | .77 | |
| 22 (numbness) | .64 | .60 | | | .60 | |
| 12 (tiring-exhausting) | .62 | .61 | | | | .05 |
| 13 (sickening) | .72 | .72 | | | | .28 |
| 14 (fearful) | .63 | .62 | | | | .60 |
| 15 (punishing-cruel) | .66 | .66 | | | | .44 |
| $\omega/\omega_S$ | | .96 | .84 | .91 | .85 | .84 |
| $\omega_H/\omega_{HS}$ | | .91 | .07 | .13 | .16 | .18 |
| $ECV/ECV_S$ | | .76 | .04 | .07 | .08 | .05 |
| PUC | | .78 | | | | |

*Note*. *N* = 552. SF-MPQ-2 = Short-Form McGill Pain Questionnaire-2. Gen. = General pain factor; Contin. = Continuous pain; Intermit. = Intermittent pain; Neuro. = Neuropathic pain; Affect. = Affective pain; $\omega$ = omega; $\omega_S$ = omega subscale; $\omega_H$ = omega hierarchical; $\omega_{HS}$ = omega hierarchical subscale; ECV = explained common variance; $ECV_S$ = explained common variance of specific factor; PUC = percentage of uncontaminated correlations.

The omega hierarchical coefficient ($\omega_H$ = .91) indicated that 91% of the variance in SF-MPQ-2 total pain scores was due to differences on the general pain factor. Conversely, omega hierarchical subscale ($\omega_{HS}$) values, which ranged from .07-.18, indicated that only 7–18% of the variance in SF-MPQ-2 subscale scores was due to differences on the corresponding specific pain factors. Dividing the obtained omega hierarchical (i.e., $\omega_H$ and $\omega_{HS}$) values by corresponding omega (i.e., $\omega$ and $\omega_S$) values revealed that 95% (i.e., .91/.96) of the reliable variance in SF-MPQ-2 total pain scores was explained by the general pain factor, whereas only 9–22% of the reliable variance in SF-MPQ-2 subscale scores was explained by the corresponding specific pain factors. These results indicate that the vast majority of the reliable variance in SF-MPQ-2 total and subscale scores was explained by the general pain factor. Thus, results suggest that SF-MPQ-2 total scores reliably measure their target construct (i.e., general pain); however, the subscale scores primarily measure the general pain construct instead of their intended specific pain constructs.

Next, we examined the explained common variance (ECV), the percentage of uncontaminated correlations (PUC), and the absolute relative parameter bias (ARPB) to determine

whether SF-MPQ-2 data (best modeled as bifactor with a strong general factor) should be specified as a unidimensional or multidimensional measurement model in SEM (Rodriguez et al., 2016b) [18]. These additional bifactor statistics enable researchers to determine whether modeling multidimensional data as unidimensional in SEM would significantly bias parameter estimates (i.e., factor loadings) (Reise et al., 2013 [46]; Rodriguez et al., 2016b [18]). The ECV reflects the proportion of total common variance explained by the general factor, and the PUC reflects the percentage of item correlations attributable only to the general factor. Based on their analyses of 50 published bifactor models, Rodriguez et al. (2016a) [19] concluded that when both the ECV and PUC are high (i.e., > .70), specifying unidimensionality in SEM is supported. According to Rodriguez et al. (2016b) [18], ARPB values reflect "the difference between an item's loading in the unidimensional solution and its general factor loading in the bifactor (i.e., truer model), divided by the general factor loading in the bifactor" (p. 145). ARPB values less than 10–15% indicate acceptable levels of parameter bias when modeling multidimensional data as unidimensional in SEM (Muthén, Kaplan, & Hollis, 1987 [47]; Rodriguez et al., 2016b [18]). As shown in Table 3, the general pain factor accounted for over three fourths of the total common variance (ECV = .76) and SF-MPQ-2 item correlations (PUC = .78). Finally, ARPB values (ranging from 0–9%, *M* = 2%) were very low across the 22 SF-MPQ-2 items, which suggests that modeling the SF-MPQ-2 as unidimensional in SEM would not result in significant parameter bias. Collectively, results of the reliability and dimensionality analyses indicated strong support for the unidimensionality of the SF-MPQ-2.

## Incremental validity

Finally, we used SEM to evaluate the incremental validity of the SF-MPQ-2 general and specific pain factors in the bifactor model by examining their unique relations with five pain-related external criteria: neck pain, depression, anxiety, stress, and symptom severity. We tested five separate structural models, one for each of the five criteria. In each model, the SF-MPQ-2 items were modeled as specified above in Model 4 (i.e., the bifactor model). We used the 10 NDI items, seven DASS-21 depression items, seven DASS-21 anxiety items, and seven DASS-21 stress items as observed indicators of neck pain, depression, anxiety, and stress latent variables, respectively, in the models predicting one of those four criteria. Finally, we used raw symptom severity scores to measure a manifest symptom severity variable in the model predicting symptom severity. In each structural model, we regressed the criterion variable (i.e., neck pain, depression, anxiety, stress, or symptom severity) onto the general and specific pain factors. Model fit was assessed using the same fit indices and cutoffs as specified above in the analyses of the SF-MPQ-2's structure.

Results revealed that all five measurement and corresponding structural models adequately fit the data (i.e., CFI ≥ .94; RMSEA ≤ .050; SRMR < .046). As shown in Table 4, the general

**Table 4. Incremental validity of SF-MPQ-2 general and specific factors (Model 4).**

| SF-MPQ-2 factor | Neck pain | Severity | Depression | Anxiety | Stress |
|---|---|---|---|---|---|
| General Pain Factor | .40*** | .20*** | .33*** | .39*** | .29*** |
| Specific Continuous Pain Factor | .13 | .05 | -.04 | -.02 | .06 |
| Specific Intermittent Pain Factor | .04 | .04 | .04 | .03 | .05 |
| Specific Neuropathic Pain Factor | .03 | -.02 | -.05 | .00 | .02 |
| Specific Affective Pain Factor | -.02 | .06 | .24*** | .29*** | .27*** |

*Note. N* = 552. SF-MPQ-2 = Short-Form McGill Pain Questionnaire-2. All values are standardized parameter estimates (*β*s).

***p < .001.

pain factor was the most consistent and strongest unique predictor of all five pain-related external criteria. The affective pain factor uniquely and positively predicted depression ($\beta$ = .24, $p < .001$), anxiety ($\beta$ = .29, $p < .001$), and stress ($\beta$ = .27, $p < .001$). The continuous, intermittent, and neuropathic pain factors did not uniquely predict any of the external criteria.

## Discussion

The purpose of this study was to evaluate the structure of the widely used SF-MPQ-2 in a large sample of adults diagnosed with Chiari malformation, a chronic health condition whose primary symptoms include pain in the head and neck region, coupled with varying degrees of fatigue and weakness [38], depression and anxiety [29], and cognitive deficits [30–33]. We compared CFA results of four different structural models: (a) an oblique four-factor model based on the SF-MPQ-2's recommended scoring (Dworkin et al., 2009) [11]; (b) a one-factor model consistent with scoring the SF-MPQ-2 for total pain; (c) a higher-order model in which the first-order factors of the oblique four-factor model mediate relations between a higher-order general pain factor and SF-MPQ-2 item responses; and (d) a bifactor model in which SF-MPQ-2 item variability is explained by both a general pain factor and specific pain factors corresponding to the four SF-MPQ-2 subscales. Results revealed the strongest support for the bifactor model, suggesting that SF-MPQ-2 item responses are due to both a general pain factor and a specific pain factor that is orthogonal to the general pain factor. Additional bifactor analyses of the SF-MPQ-2's model-based reliability and dimensionality revealed that most of the SF-MPQ-2's reliable variance is explained by a general pain factor, and that the instrument can be modeled unidimensionally and scored as a general pain measure. Results also indicated that the general and affective pain factors in the bifactor model uniquely predicted pain-related external criteria (e.g., depression, anxiety, and stress); however, the continuous, intermittent, and neuropathic factors did not. Following is a more detailed discussion of the major findings.

### SF-MPQ-2 structure

CFA results indicated that, of the four models tested, only the bifactor model adequately fit the SF-MPQ-2 data in our sample of adults with Chiari malformation. Moreover, the bifactor model provided a significantly better fit than did the higher-order model. These results suggest that variability in SF-MPQ-2 item responses is attributable to a combination of a general pain construct and specific pain constructs. These findings call into question earlier findings regarding the structure of the SF-MPQ-2. For example, several previous CFA studies reported support for an oblique four-factor model (e.g., Gauthier et al., 2014 [14]; Lovejoy et al., 2012 [15]) or for interpreting the four SF-MPQ-2 subscales (Dworkin et al., 2009 [11]; Dworkin et al., 2015 [13]). However, the oblique four-factor solution in two of those studies (Gauthier et al., 2014 [14]; Lovejoy et al., 2012 [15]) did not meet conventional fit index cutoffs for adequate fit (e.g., CFI close to .95, RMSEA ≤ .06; Kline, 2016 [17]). Furthermore, Dworkin and colleagues [11,13] conducted separate CFAs on SF-MPQ-2 item sets composing a given subscale; therefore, it is unclear from their analyses to what extent the items that clustered to reflect a narrower pain construct also (or instead) were influenced by a general source of variance (i.e., total pain).

An alternative to the oblique four-factor model—the higher-order model—also has received support in previous research. Lovejoy et al. (2012) [15] found that a higher-order model fit the SF-MPQ-2 data as well as an oblique four-factor model in their sample of 186 veterans with chronic pain symptoms. However, neither model met recommended cutoffs for adequate fit (e.g., CFI close to .95, RMSEA ≤ .06; Kline, 2016) [17]. Consistent with Lovejoy et al.'s (2012) findings [15], the higher-order model fit the current data as well as the oblique

four-factor model did; however, neither model provided an adequate fit using recommended cutoffs.

Support for a bifactor model implies that, in addition to a general factor influencing item responses, each subset of items corresponding to a given subscale loads substantively on (and thus is well defined by) the corresponding specific factor (Brunner et al., 2011) [16]. In the current bifactor model, however, only 12 of 22 SF-MPQ-2 items loaded significantly on a specific pain factor, and only eight of the 12 significant loadings exceeded .30. In contrast, all 22 items loaded significantly and substantively (ranging from .47 to .75) on the general pain factor. Further, only one of the 22 SF-MPQ-2 items (item 21: "tingling") loaded more strongly on a specific pain factor than on the general pain factor. Thus, although a multidimensional bifactor model best captured the structure of the SF-MPQ-2 in our sample, variability in item responses was primarily driven by the general pain factor.

### Bifactor reliability and dimensionality

Consistent with recent recommendations for bifactor modeling (Rodriguez et al., 2016a, 2016b) [18,19], we calculated a number of additional bifactor statistics to evaluate the model-based reliability and dimensionality of the SF-MPQ-2. Bifactor reliability analyses revealed that the specific SF-MPQ-2 subscales evidenced unacceptably low reliabilities after accounting for the variance attributable to the general pain factor (Rodriguez et al., 2016b) [18]. In contrast, total pain scores demonstrated a high level of reliability, suggesting that SF-MPQ-2 total scores are mostly attributable to a general pain factor. These results support the common practice of scoring and interpreting the SF-MPQ-2 for total pain (e.g., Dworkin et al., 2015 [13]; Gauthier et al., 2104 [14]; Lovejoy et al., 2012 [15]). However, because SF-MPQ-2 raw subscale scores primarily recapture the general pain factor instead of their intended specific pain constructs, interpreting the raw subscale scores as measuring unique pain constructs (i.e., beyond general pain) is not recommended.

As further evidence of the instrument's unidimensionality, additional bifactor indices revealed that over three-fourths of the explained common variance and correlations among the 22 SF-MPQ-2 items was attributable only to the general pain factor. Furthermore, although the one-factor model provided a poor fit to the data, absolute relative parameter bias values indicated that item factor loadings in the unidimensional solution and the general pain factor of the bifactor solution were nearly identical. This finding suggests that although the SF-MPQ-2 was found to be a multidimensional measure (i.e., best fitting a bifactor model), modeling the SF-MPQ-2 items as unidimensional in SEM would not bias the instrument's ability to capture the general pain construct of the truer (but more complex) bifactor model (Rodriguez et al., 2016b [18]).

### Incremental validity

Although our results do not support the practice of using raw SF-MPQ-2 subscale scores to measure specific subdomains of pain, modeling the SF-MPQ-2 as a bifactor solution (i.e., orthogonal general and specific pain factors) in SEM revealed unique associations of general and specific pain factors with five pain-related criteria. The general pain factor uniquely and positively predicted neck pain, symptom severity, depression, anxiety, and stress. Beyond general pain, the only other unique predictor of the external criteria was affective pain, which uniquely predicted depression, anxiety, and stress. Because affective pain "refers to how unpleasant or disturbing the pain feels" (Fillingim et al., 2016, p. T11 [7]), and past research has reported an increase in anxiety and depression in individuals diagnosed with CM (Garcia et al., 2019) [29], it is hardly surprising that participants' experience of affective pain accounted

for unique variance in their experience of depression, anxiety, and stress, all of which involve some degree of emotional discomfort. These findings are somewhat consistent with previous studies relating raw SF-MPQ-2 subscale scores to measures of depression and anxiety (Dworkin et al., 2015 [13]; Gauthier et al., 2014 [14]; Lovejoy et al., 2012 [15]). In those studies, affective pain related to depression and anxiety only slightly more so than did the three sensory pain constructs. It seems likely that the modest differential relations of specific pain constructs with depression and anxiety found in previous studies were due to the raw SF-MPQ-2 subscale scores, which primarily remeasure general pain instead of their intended specific pain constructs. In sum, results of incremental validity analyses suggest that, although scoring the SF-MPQ-2 raw subscales is contraindicated, specifying a bifactor latent model in SEM with general and specific pain constructs can allow researchers to determine the unique relations of these constructs—most notably affective pain—to external criteria.

## Implications for future research and clinical assessment

Our results of an optimal bifactor model with a general and a specific factor applied to Chiari patients have implications for both research and application of the assessment of pain using the SF-MPQ-2. Conceptually, the pattern of findings here might point to a general factor attributable to the acute experience of pain associated with tissue damage, encroachment, or inflammation (i.e., nociceptive), whereas the more specific pain factor (i.e., affective) may reflect more chronic centralized pain (e.g., Allen et al., 2018 [20]; 2022 [25]). Evidence supporting this dichotomy may be found in the pattern of less optimal post-surgical improvement observed among Chiari patients when there is greater than a two-year interval between initial diagnosis and surgery (Labuda et al., 2022 [48]). Such a time period might permit increased central sensitization of pain to occur [20], quite possibly leading to increased activation of emotive-affective networks. Also, the present incremental validity analyses underscore the relationship between the specific (affective) factor and the DASS21 measures of Depression, Anxiety, and Stress–variables which are routinely associated with chronic pain [29] (Garcia et al., 2019). It is also worth noting that Gholampour and Taher (2018) [49] and Gholampour and Gholampour (2020) [50] have implicated cerebrospinal fluid (CSF) pressure with degree of headache pain in Chiari malformation patients (as measured by phase contrast MRI, or cine-flow). In support of this, Garcia et al. (2022) [25] found that a smaller anterior cervical CSF space (between C2 and the foramen magnum in the upper spine) was associated with increased pain in Chiari patients as adult age increased. As such, future research should assess whether CSF pressure in Chiari patients is differentially related to acute (e.g., Val Salva) and chronic (i.e., centrally sensitized) pain.

Psychometrically, our findings also suggest that researchers can measure the SF-MPQ-2's general pain factor using either a bifactor model or a more parsimonious unidimensional measurement model in SEM. As such, our findings support the common practice (among researchers and clinicians) of calculating and interpreting raw SF-MPQ-2 total mean scores. Conversely, given that very little reliable variance was explained by specific subscale scores here, there is questionable support for using raw SF-MPQ-2 subscale scores in either a research or clinical context. Obviously, such a finding is anticipated to raise questions and resistance, particularly among those who have used this measure. Importantly, the intent is not to undermine the utility of the instrument; rather, to better understand how the SF-MPQ-2 captures the overall experience of pain, and to align the measure with the physical, qualitative, and affective dimensions of pain.

We recognize that our results must be considered in light of some important caveats. First and foremost, our sample consisted of adults diagnosed with Chiari malformation, a fairly rare

chronic health condition characterized by varying degrees of sensory, motor, cognitive and psychosocial complications, with a higher likelihood of head and neck pain (Houston et al., 2022 [51]; Fischbein et al., 2015 [38]). As such, we cautiously underscore our reported findings, cognizant of the need to further assess the psychometric properties of the SF-MPQ-2 within the broader experience of self-reported pain across diverse medical conditions. It is noteworthy, however, that in terms of levels of self-reported pain, our sample was quite comparable (i.e., within roughly one-half standard deviation) to other chronic pain samples reported in the literature (e.g., Dworkin et al., 2009 [11]; Gauthier et al., 2014 [14]; Lovejoy et al., 2012 [15]). Nevertheless, replication with other samples representing one or more chronic (e.g., fibromyalgia, advanced cancer, arthritis) or acute (e.g., acute low back pain) pain conditions is warranted. In addition, the vast majority of our participants identified as female and white, thus somewhat limiting the implications here. Inasmuch as women and men can differ in their subjective experience of pain (e.g., Etherton, Lawson, & Graham, 2014 [52]), it remains an important research consideration to evaluate gender effects on psychometric aspects of subjective pain measures. Future researchers are encouraged to replicate the current study with further attention to gender differences and sample diversity.

## Conclusion

Collectively, our results provide strong support for calculating and interpreting raw SF-MPQ-2 total scores for a general pain construct. Furthermore, because SF-MPQ-2 item responses largely reflect a general pain construct, modeling the instrument as a unidimensional measurement model in SEM is justified. Nevertheless, there may be instances when researchers wish to examine the predictive utility of both the general and specific pain constructs. In such instances, researchers should choose to represent the SF-MPQ-2 multidimensionally, i.e., as a bifactor model. Finally, our results contraindicate the common practice of calculating and interpreting the raw SF-MPQ-2 subscale scores for specific pain constructs.

## Supporting information

**S1 File.**
(SAV)

## Author Contributions

**Conceptualization:** David M. Tokar, Kevin P. Kaut, Philip A. Allen.

**Formal analysis:** David M. Tokar, Philip A. Allen.

**Project administration:** David M. Tokar.

**Writing – original draft:** David M. Tokar, Kevin P. Kaut, Philip A. Allen.

**Writing – review & editing:** David M. Tokar, Kevin P. Kaut, Philip A. Allen.

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
