## [Decision Letter · Decision Letter 0]

2 May 2023

PONE-D-22-29795Revisiting the Factor Structure of the Short-Form McGill Pain Questionnaire-2 (SF-MPQ-2): Evidence for a Bifactor Model in Individuals with Chiari MalformationPLOS ONE

Dear Dr. Allen,

Thank you for submitting your manuscript to PLOS ONE. After careful consideration, we feel that it has merit but does not fully meet PLOS ONE’s publication criteria as it currently stands. Therefore, we invite you to submit a revised version of the manuscript that addresses the points raised during the review process.

Update the literature review with more recent literature. Please also complete the following minor amendments:  1. “An estimated 80-100 million American adults experience some form of chronic pain and that estimate will undoubtedly increase as the population continues to age”. It will be better if the authors can use a newer reference. These statistics may have been recently updated. In addition, all the studies that were reviewed in the introduction section are from before 2017. It is necessary to mention the newer papers from the last three years, as well.

2. It can be better if you present your personal hypotheses and reasons (from the clinical point of view) for your results in the discussion section. For example, your results showed the strongest support for the bifactor model, suggesting that SF-MPQ-2 item responses are due to both a general pain factor and a specific pain factor that is orthogonal to the general pain factor. This is a valuable find. In addition to the results of your statistical analysis, what do you think is the reason for this from the clinical point of view? You can discuss it to open new windows for future research.

3. Previous studies showed a relationship between CSF dynamic parameters and clinical signs such as pain [10.1016/j.wneu.2018.05.108] [10.1038/s41598-020-72961-0]. Do you think that future studies can consider this as a factor to develop the questionnaire?

Please also correct these typo errors:

1. “… Like Dworkin et al. (2009), Dworkin et al. (2105) performed separate…”. Correct “(2105)”

2. “Missing Completely at Random (MCAR)” should be changed to “missing completely at random (MCAR)”.

3. Write the full SEM form "Structural Equation Modeling" the first time you used it in the manuscript (page 13), not page 14.

We look forward to receiving your revised manuscript.

Kind regards,

Leica S. Claydon-Mueller

Academic Editor

PLOS ONE

2. Please provide additional details regarding ethical approval in the body of your manuscript. In the Methods section, please ensure that you have specified the name of the IRB/ethics committee that approved your study.

Additional Editor Comments (if provided):

Thank you for submitting your manuscript to PLOS ONE. Sincere apologies for the delay in reviewing we invited close to 20 reviewers. The statistician's review indicates the statistics are sound.

The decision is Minor Revisions. Please update the introduction with more recent literature. Please also complete these minor modifications:

1. “An estimated 80-100 million American adults experience some form of chronic pain and that estimate will undoubtedly increase as the population continues to age”. It will be better if the authors can use a newer reference. These statistics may have been recently updated. In addition, all the studies that were reviewed in the introduction section are from before 2017. It is necessary to mention the newer papers from the last three years, as well.

2. It can be better if you present your personal hypotheses and reasons (from the clinical point of view) for your results in the discussion section. For example, your results showed the strongest support for the bifactor model, suggesting that SF-MPQ-2 item responses are due to both a general pain factor and a specific pain factor that is orthogonal to the general pain factor. This is a valuable find. In addition to the results of your statistical analysis, what do you think is the reason for this from the clinical point of view? You can discuss it to open new windows for future research.

3. Previous studies showed a relationship between CSF dynamic parameters and clinical signs such as pain [10.1016/j.wneu.2018.05.108] [10.1038/s41598-020-72961-0]. Do you think that future studies can consider this as a factor to develop the questionnaire?

Typo errors:

1. “… Like Dworkin et al. (2009), Dworkin et al. (2105) performed separate…”. Correct “(2105)”

2. “Missing Completely at Random (MCAR)” should be changed to “missing completely at random (MCAR)”.

3. Write the full SEM form "Structural Equation Modeling" the first time you used it in the manuscript (page 13), not page 14.

Kindest regards,

Leica

Reviewers' comments:

Reviewer's Responses to Questions

**Comments to the Author**

1. Is the manuscript technically sound, and do the data support the conclusions?

Reviewer #1: Yes

Reviewer #2: Yes

2. Has the statistical analysis been performed appropriately and rigorously? 

Reviewer #1: Yes

Reviewer #2: Yes

3. Have the authors made all data underlying the findings in their manuscript fully available?

Reviewer #1: Yes

Reviewer #2: Yes

4. Is the manuscript presented in an intelligible fashion and written in standard English?

Reviewer #1: Yes

Reviewer #2: Yes

5. Review Comments to the Author

Reviewer #1: Chiari malformation is a complicated disorder and pain is one of the controversial challenges related to this disorder. Because pain is a complex and multifactor phenomenon. Despite extensive research and study, there is still much that is not fully understood about it. So, revisiting the pain questionnaire can be really interesting for the readers of PloS One. The authors strived to evaluate the factor structure of the SF-MPQ-2 to measure the multidimensional qualities of pain and total pain. They evaluated the fit of four competing measurement models in 552 adults (73 cases contained missing item-level data points) diagnosed with Chiari malformation. They hypothesized that continuous, intermittent, neuropathic, effective, and general pain factor scores will relate uniquely and positively with anxiety, depression, and stress. The authors showed that most of the SF-MPQ-2’s reliable variance is explained by a general pain factor and that the instrument can be modeled unidimensionally and scored as a general pain measure. Their results also showed that the general and affective pain factors in the bifactor model uniquely predicted pain-related external criteria but the continuous, intermittent, and neuropathic factors did not. This study is well-designed, and the method and result sections are quite clear for the readers. The introduction section should be updated with some newer papers. Taken together, this finding provides strong support for calculating and interpreting raw SF-MPQ-2 total scores for a general pain to construct. Hence, I strongly believe that this study provided important practical data for neurosurgeons and has good potential for future citations. So, I suggest publishing this paper after considering the following minor comments.

1. “An estimated 80-100 million American adults experience some form of chronic pain and that estimate will undoubtedly increase as the population continues to age”. It will be better if the authors can use a newer reference. These statistics may have been recently updated. In addition, all the studies that were reviewed in the introduction section are from before 2017. It is necessary to mention the newer papers from the last three years, as well.

2. It can be better if you present your personal hypotheses and reasons (from the clinical point of view) for your results in the discussion section. For example, your results showed the strongest support for the bifactor model, suggesting that SF-MPQ-2 item responses are due to both a general pain factor and a specific pain factor that is orthogonal to the general pain factor. This is a valuable find. In addition to the results of your statistical analysis, what do you think is the reason for this from the clinical point of view? You can discuss it to open new windows for future research.

3. Previous studies showed a relationship between CSF dynamic parameters and clinical signs such as pain [10.1016/j.wneu.2018.05.108] [10.1038/s41598-020-72961-0]. Do you think that future studies can consider this as a factor to develop the questionnaire?

Typo errors:

1. “… Like Dworkin et al. (2009), Dworkin et al. (2105) performed separate…”. Correct “(2105)”

2. “Missing Completely at Random (MCAR)” should be changed to “missing completely at random (MCAR)”.

3. Write the full SEM form "Structural Equation Modeling" the first time you used it in the manuscript (page 13), not page 14.

Reviewer #2: The paper is well-written and structured, and the statistical analysis is well-conducted in all of its parts/aspects (description of the data, descriptive preliminaries, main analysis, model selection, and validation). The reference to the literature for each statement/choice made, and the use of more sophisticated statistical tools when necessary, are aspects that deserve to be appreciated. As such, I suggest to accept the paper as it is.

6. PLOS authors have the option to publish the peer review history of their article (what does this mean?). If published, this will include your full peer review and any attached files.

Reviewer #1: **Yes: **Seifollah Gholampour

Reviewer #2: No

---

## [Author Response · Author response to Decision Letter 0]

24 May 2023

Dear Dr. Claydon-Mueller:

 We have now revised “Revisiting the Factor Structure of the Short-Form McGill Pain Questionnaire-2 (SF-MPQ-2): Evidence for a Bifactor Model in Individuals with Chiari Malformation” as you and the Reviewer recommended. In particular, we added a more recent citation of the incidence of chronic pain in the US (a 2022 PAIN paper noted at the bottom of this cover letter) and we added our hypotheses regarding the interpretation of the general and specific bifactor constructs (acute Val Salva pain and chronic centrally sensitized pain). We also added the recommended references and corrected the noted typographical errors. Finally, we added a new paragraph in the Method section entitled “Ethical Statement” and have now added our dataset in your submission portal.

Regards,

 Phil Allen

PONE-D-22-29795

Revisiting the Factor Structure of the Short-Form McGill Pain Questionnaire-2 (SF-MPQ-2): Evidence for a Bifactor Model in Individuals with Chiari Malformation

PLOS ONE

Dear Dr. Allen,

Thank you for submitting your manuscript to PLOS ONE. After careful consideration, we feel that it has merit but does not fully meet PLOS ONE’s publication criteria as it currently stands. Therefore, we invite you to submit a revised version of the manuscript that addresses the points raised during the review process.

Update the literature review with more recent literature. Please also complete the following minor amendments: 

1. “An estimated 80-100 million American adults experience some form of chronic pain and that estimate will undoubtedly increase as the population continues to age”. It will be better if the authors can use a newer reference. These statistics may have been recently updated. In addition, all the studies that were reviewed in the introduction section are from before 2017. It is necessary to mention the newer papers from the last three years, as well.

Response: We have now updated the chronic pain portion at the beginning of our manuscript with the Yong, Mullins and Bhattacharyya (2022, PAIN) paper from the 2019 National Health Interview Survey (based on a very large sample).

2. It can be better if you present your personal hypotheses and reasons (from the clinical point of view) for your results in the discussion section. For example, your results showed the strongest support for the bifactor model, suggesting that SF-MPQ-2 item responses are due to both a general pain factor and a specific pain factor that is orthogonal to the general pain factor. This is a valuable find. In addition to the results of your statistical analysis, what do you think is the reason for this from the clinical point of view? You can discuss it to open new windows for future research.

Response: On p. 24, paragraph 2 we now present our hypothesis of why this bifactor pattern with a general factor and on specific factor (related to the Affective sub-factor) occurred. We suspect that the general factor is related to nociceptive (acute) pain associated with head and neck tissue damage in Chiari—which we agree with this reviewer is likely related to CSF pressure effects that affect tissue compliance at the cranio-cervical junction. Alternatively, we suspect that the specific (affective) factor is related to central sensitization of pain (a chronic effect). Our finding of this specific factor being significantly related to depression, anxiety, and stress in the incremental validity analysis is consistent with results from Garcia et al. (2019—Chiari pain is related to depression, anxiety and stress—as well as trauma scores) and Allen et al. (2018—Chiari pain appears to be related to central sensitization of pain). 

3. Previous studies showed a relationship between CSF dynamic parameters and clinical signs such as pain [10.1016/j.wneu.2018.05.108] [10.1038/s41598-020-72961-0]. Do you think that future studies can consider this as a factor to develop the questionnaire?

Response: We have added these two recommended references showing the importance of CSF pressure on headache pain in Chiari, as well as linking this to Garcia et al. (2022—who found that anterior CSF space in the cervical spine area mediated age differences in pain in Chiari patients). We believe that it will be important to determine in future research if this Affective specific factor remains after surgery but the general factor is attenuated.

Please also correct these typo errors:

1. “… Like Dworkin et al. (2009), Dworkin et al. (2105) performed separate…”. Correct “(2105)”

Response: We changed the 2105 date to 2015.

2. “Missing Completely at Random (MCAR)” should be changed to “missing completely at random (MCAR)”.

Response: We made this recommended revision in the text. 

3. Write the full SEM form "Structural Equation Modeling" the first time you used it in the manuscript (page 13), not page 14.

Response: We spelled out structural equation modeling on the first mention (and linked it to SEM). 

5. 2. Please provide additional details regarding ethical approval in the body of your manuscript. In the Methods section, please ensure that you have specified the name of the IRB/ethics committee that approved your study.

Response: On p. 9, we have now added an ethics statement (that also notes that all participants provided electronic informed consent).

Response: We obtained electronic written informed consent—and all participants were adults. 

Response: Our study used information from the Chiari 1000 sample in which Chiari patients completed a comprehensive web survey (including the SF-MPQ-2, the DASS21, and the Oswestry disability scale used in our analyses). 

Response: Our new funding statement is: This research was supported by a Conquer Chiari Research Grant from the Conquer Chiari Foundation as well as NIH grant 1R15NS109957-01A1.

Response: The Conquer Chiari grant does not include a grant number—but we now do include the NIH R15 grant number.

Response: We included our data as a support file (in SPSS form) for Dryad. This is what technical support recommended.

N/A

It will be posted in Dryad.

The link is: Dryad.

We added five new References (no retracted articles were included):

Allen, P.A., Houston, J.R., Pollock, J.W., Buzzelli, C., Li, X., Harrington, A..C., Martin, B.A., Loth, F., Lien, M.-C., Maleki, J., & Luciano, M.G. (2014). Task-specific and general cognitive effects in Chiari malformation Type I. PLoS ONE, 9(4),e94844.

Gholampour, S. & Taher, M. (2018). Relationship of Morphologic Changes in the Brain and Spinal Cord and Disease Symptoms with Cerebrospinal Fluid Hydrodynamic Changes in Patients with Chiari Malformation Type I. World Neurosurgery, 116, e830-e839. https://doi.org/10.1016/j.wneu.2018.05.108

Gholampour, S., & Gholampour, H. (2020). Correlation of a new hydrodynamic index with other efective indexes in Chiari I malformation patients with different associations. Scientific Reports, 10, Article number: 15907. https://doi.org/10.1038/s41598-020-72961-0

Houston, J.R., Maleki, J., Loth, F., Klinge, P.M., Allen, P.A. (2022). Influence of pain on cognitive dysfunction and emotion dysregulation in Chiari malformation Type I. In M. Adamaszek, M. Manto, and D. Schutter (Eds.), The emotional cerebellum. pp 155-178. Advances in Experimental Medicine and Biology 1378, https://doi.org/10.1007/978-3-030-99550-8_11

Yong, R.J., Mullins, P.M., & Bhattacharyya. (2022). Prevalence of chronic pain among adults in the United States. PAIN, 163(2), e328-e332. Doi:10.1097/j.pain.0000000000002291

---

## [Editor Report · Decision Letter 1]

1 Jun 2023

Revisiting the Factor Structure of the Short-Form McGill Pain Questionnaire-2 (SF-MPQ-2): Evidence for a Bifactor Model in Individuals with Chiari Malformation

PONE-D-22-29795R1

Dear Dr. Allen,

We’re pleased to inform you that your manuscript has been judged scientifically suitable for publication and will be formally accepted for publication once it meets all outstanding technical requirements.

Kind regards,

Leica S. Claydon-Mueller

Academic Editor

PLOS ONE

---

## [Editor Report · Acceptance letter]

15 Jun 2023

PONE-D-22-29795R1 

Revisiting the Factor Structure of the Short-Form McGill Pain Questionnaire-2 (SF-MPQ-2): Evidence for a Bifactor Model in Individuals with Chiari Malformation 

Dear Dr. Allen:

I'm pleased to inform you that your manuscript has been deemed suitable for publication in PLOS ONE. Congratulations! Your manuscript is now with our production department. 

Kind regards, 

on behalf of

Dr. Leica S. Claydon-Mueller 

Academic Editor

PLOS ONE